# The Trigeminal Sensory System and Orofacial Pain

**DOI:** 10.3390/ijms252011306

**Published:** 2024-10-21

**Authors:** Hyung Kyu Kim, Ki-myung Chung, Juping Xing, Hee Young Kim, Dong-ho Youn

**Affiliations:** 1Department of Physiology, Yonsei University College of Medicine, Seoul 03722, Republic of Korea; badawanabi@gmail.com (H.K.K.); xingjuping@yuhs.ac (J.X.); 2Department of Oral Physiology, School of Dentistry, Kyungpook National University, Daegu 41940, Republic of Korea; 3Department of Physiology and Neuroscience, College of Dentistry, Gangneung-Wonju National University, Gangneung 25457, Republic of Korea; ckm@gwnu.ac.kr

**Keywords:** trigeminal ganglion, trigeminal sensory nuclei, mesencephalic nucleus, principal nucleus, spinal trigeminal nucleus, paratrigeminal nucleus, synaptic transmission, brainstem parasympathetic system, orofacial neuropathic pain, trigeminal neuralgia, headache, burning mouth syndrome

## Abstract

The trigeminal sensory system consists of the trigeminal nerve, the trigeminal ganglion, and the trigeminal sensory nuclei (the mesencephalic nucleus, the principal nucleus, the spinal trigeminal nucleus, and several smaller nuclei). Various sensory signals carried by the trigeminal nerve from the orofacial area travel into the trigeminal sensory system, where they are processed into integrated sensory information that is relayed to higher sensory brain areas. Thus, knowledge of the trigeminal sensory system is essential for comprehending orofacial pain. This review elucidates the individual nuclei that comprise the trigeminal sensory system and their synaptic transmission. Additionally, it discusses four types of orofacial pain and their relationship to the system. Consequently, this review aims to enhance the understanding of the mechanisms underlying orofacial pain.

## 1. Introduction

The trigeminal sensory system consists of the trigeminal nerve, the trigeminal ganglion, and the trigeminal sensory nuclei. Furthermore, the trigeminal sensory nuclei contain several brainstem nuclei, including the mesencephalic nucleus in the midbrain, the principal nucleus in the pons, and the spinal trigeminal nucleus, along with other smaller nuclei primarily located in the pons, medulla, and spinal cord. This system processes sensory information, including proprioception, touch, temperature, and pain, from the orofacial region, particularly the teeth and their surrounding structures, masticatory muscles, cornea, and the trigeminal nerve territory of the facial skin. In addition to the specific carriers of sensory modalities, all information is subject to modulation and integration by the trigeminal sensory system. Therefore, a detailed understanding of the cellular composition of individual nuclei, intra- and internuclear synaptic connections, and extra-trigeminal influences, such as brainstem parasympathetic flow, on the system can serve the fundamental understandings of orofacial somatic sensation, including pain. In this review, we aim to explain the cellular organization, connections, and synaptic transmission of the trigeminal sensory system. Additionally, we discuss four distinct types of orofacial pain and their relationship to this system.

## 2. Organization of the Trigeminal Sensory System

### 2.1. Trigeminal Ganglion

The trigeminal (semilunar or Gasserian) ganglion is located in the middle cranial fossa at the base of the skull. The neurons within this ganglion receive sensory information via the trigeminal nerve from the skin, mucous membranes, teeth, masticatory muscles (~10%; proprioceptors), Golgi tendon organs, and temporomandibular joint. Then, they transmit the information to the brainstem sensory nuclei in the central nervous system (CNS) [1]. The sensory submodalities transmitted by the trigeminal ganglion neurons include pain, temperature, pressure, and touch sensations.

Morphologically, the ganglion neurons are pseudounipolar and are surrounded by satellite cells, which are the protective glial cells in the ganglion. Trigeminal ganglion neurons can be classified into large/light (type A) cells, smaller/dark (type B) cells, and small (type C) cells [2]. This classification approximately correlates with afferent distribution. For instance, primary afferents innervating the tooth pulp in the rat primarily connect to type A cells, while those innervating the cornea connect to type B cells [3]. Nearly half of the neurons contain calcitonin gene-related peptide (CGRP), approximately 15% contain substance P, and some contain both peptides [4,5]. A discrete population of CGRP- or substance P-immunoreactive neurons co-localized with neurons expressing trkA, trkB, and trkC receptors for the neurotrophins [6] and those expressing vanilloid receptor 1 (VR1) receptors [7,8], which is now called transient receptor potential (TRP) cation channel subfamily V member 1 (i.e., TRPV1). In addition to those peptides, 32–45% of rat trigeminal ganglion neurons contain glutamate [9,10]. Glutamatergic terminals can be found throughout the rodent trigeminal sensory complex [11]. Interestingly, trigeminal ganglion neurons are surrounded by various fibers including noradrenergic sympathetic axons and vasoactive intestinal peptide (VIP)-positive parasympathetic fibers [12], suggesting a potential modulation of ganglion transmission by autonomic influences.

The peripheral nerves originating from the trigeminal ganglion divide into three main branches: ophthalmic (V1), maxillary (V2), and mandibular (V3) branches. The V1 branch exits the skull through the superior orbital fissure, and innervates the forehead, scalp, upper eyelid, and conjunctiva of the eye. The V2 branch exits via the foramen rotundum and traverses the pterygopalatine fossa, providing innervation of the cheek, upper lip, nostril, upper jaw, and teeth. The V3 branch exits through the foramen ovale and the infratemporal fossa, and innervates the lower jaw and teeth, lower lip, cheek, and the anterior two-thirds of the tongue. Additionally, fibers from the autonomic nervous system often join the trigeminal nerve branches to reach peripheral tissues, including salivary and sweat glands [13] (Figure 1). Parasympathetic fibers via the ciliary ganglion, which come through the oculomotor nerve (cranial nerve III) from the Edinger–Westphal nucleus in the midbrain, join the nasociliary nerve of the ophthalmic branch and innervate the pupillary sphincter and the ciliary muscle of the eye. Parasympathetic fibers via the pterygopalatine ganglion, which come through the greater petrosal nerve of the facial nerve (cranial nerve VII) from the superior salivatory nucleus in the pons, join the zygomatic nerve of the maxillary branch. They subsequently join the lacrimal nerve of ophthalmic branch and innervate the lacrimal gland. Furthermore, parasympathetic fibers via the otic ganglion, which come through the tympanic nerve and the lesser petrosal nerve of the glossopharyngeal nerve (cranial nerve IX) from the inferior salivatory nucleus in the medulla, join the auriculotemporal nerve of the mandibular branch and innervate the parotid gland. Parasympathetic fibers via the submandibular ganglion, that come through the chorda tympani branch of the facial nerve, join the lingual nerve of the mandibular branch and innervate the sublingual and submandibular glands (Figure 1).

### 2.2. Trigeminal Sensory Nuclei

The trigeminal sensory nuclei are a collection of nuclei located in the brainstem responsible for processing sensory information from the face and head. Each subnucleus within the trigeminal sensory nuclei is specialized for processing different types of sensory information, including light touch, pressure, pain, and temperature. The information processed in the trigeminal sensory nuclei is transmitted to other regions of the central nervous system, contributing to the overall perception of sensory stimuli from the face and head. The trigeminal sensory nuclei consist of the mesencephalic nucleus (Me5), the main or principal sensory nucleus (Pr5), the spinal trigeminal nucleus (Sp5), and smaller nuclei such as the paratrigeminal nucleus (Pa5) and the peritrigeminal nucleus [1]. However, the connections and functions of the peritrigeminal nucleus are unknown, and therefore this review does not address this particular nucleus.

#### 2.2.1. The Mesencephalic Nucleus

The Me5 comprises a band of scattered neurons that extends from the level of Pr5 or the trigeminal motor nucleus in the pons caudally to the lateral margin of the periaqueductal gray in the midbrain rostrally. This nucleus contains the somata of primary afferents from masticatory or extraocular muscle spindles, as well as some periodontal receptors of both maxillary and mandibular teeth [12]. The majority of neurons in this region are large pseudounipolar cells; however, small-diameter cells with multipolar processes also exist, and both neurons receive glutamatergic inputs [14]. Me5 neurons primarily contain glutamate, and, unlike trigeminal ganglion cells, do not express neuropeptides such as CGRP and substance P [12]. Interestingly, Me5 cells exhibit membrane oscillations, which may result from gap junctions and electrical coupling between them [15]. In addition to their central axons projecting to trigeminal motor nuclei and hypoglossal nuclei, Me5 neurons also project to sensory regions such as the dorsomedial part of the Pr5 and Sp5, as well as other nuclei, including the nucleus tractus solitarius and the supratrigeminal region [16] (Figure 2). This anatomical organization suggests that the Me5 plays a role in the integration of jaw movements and proprioception during mastication [1]. Furthermore, its extra-trigeminal connections to the hypothalamus suggest a potential influence on satiety through dopaminergic modulation of chewing speed [17].

#### 2.2.2. The Principal Sensory Nucleus

The Pr5 is one of the two main nuclei in the trigeminal sensory nuclei system located in the brainstem. It is responsible for processing information related to touch, pressure, and temperature sensations from the face, which are conveyed by the trigeminal nerve. The Pr5 receives sensory information from the trigeminal nerve and integrates this information with other sensory inputs to create a comprehensive representation of the sensory experience. In both the rats and cats, the Pr5 contains synaptic glomeruli, a complex synaptic ultrastructure in which a core axonal bouton is surrounded by multiple dendrites and axonal boutons. These synaptic glomeruli are believed to play a significant role in integrating various sensory information. Animal studies have demonstrated that neurons in the principal nucleus are mechanoreceptive, exhibiting low thresholds and small receptive fields and providing discriminative tactile sensations from the face [18]. Consequently, this nucleus is comparable to the dorsal column nuclei. Once the sensory information has been processed, the Pr5 transmits it, via numerous projection neurons, to higher brain regions, such as the ventroposterior medial nucleus (VPM) and, to a lesser extent, the posterior complex (Po) of the contralateral thalamus [1], for further processing and integration (Figure 2). From a chemical neuroanatomy perspective, 60–70% of the projection neurons in the rodent principal nucleus contain glutamate [19], and many interneurons are immunoreactive for γ-aminobutyric acid (GABA) [1].

#### 2.2.3. The Spinal Trigeminal Nucleus

On the other hand, another nucleus within the trigeminal sensory nuclei, known as the Sp5, is divided into three subnuclei: oralis (Sp5O), interpolaris (Sp5I), and caudalis (Sp5C). Although all three subnuclei are involved in processing sensory information from the face and head, they serve distinct functions and receive input from different sensory neurons. The Sp5O is primarily responsible for processing touch and pressure sensations from the oral cavity. It contains small (12–17 μm) oval cells and medium-sized oval, multipolar, or fusiform cells (25–30 μm long and ~10 μm wide) [20]. Many neurons in this subnucleus are glutamatergic [21] and also GABAergic [22]. The Sp5O has been shown to receive extensive intraoral information [23,24], supported by the observation that vascular lesions in this area can lead to a loss of oral sensation in humans [25]. However, it is also true that responses in this area exhibit widespread receptive fields, demonstrating modality convergence from both cutaneous and tooth receptors [20]. Additionally, Sp5O neurons project to the Me5, facial, and hypoglossal nuclei, with some projections extending to the contralateral facial nucleus [1] (Figure 2). There are also projections to the cervical spinal cord, the thalamic Po, and the tectum, the zona incerta, the anterior pretectal nuclei, Pr5, and other Sp5 subnuclei [20,26].

The Sp5I contains a heterogeneous population of neurons, comprising many small to medium-sized (15–20 μm) and oval or spindle-shaped neurons, as well as some large, scattered neurons (30–40 μm) [20]. It receives widespread inputs from intraoral structures, including tooth pulp, although these inputs are generally less dense than those for Sp5O [23,24]. The neurons are responsive to both low-threshold mechanoreceptors and nociceptors in the skin and periodontium [27]. In the rats, Sp5I neurons are either glutamatergic or GABAergic [19,21,22]. The Sp5I neurons projects to the thalamus, cerebellum, superior colliculus, and spinal cord [28,29,30,31] (Figure 2). In the rats, there are thalamic projections of Sp5I to both VPM and the medial part of Po [32]. Additionally, a study suggests that an area near the Sp5I/C border may correspond to a human sneezing center [33].

The Sp5C is the most caudal (posterior) and functionally complex of the three subnuclei. It is also referred to as the medullary dorsal horn due to its similar laminar organization to the dorsal horn of the spinal cord [34]. This region is primarily involved in processing noxious or painful stimuli, and plays a crucial role in the transmission and modulation of pain signals. Additionally, it is involved in regulating autonomic and emotional responses to pain. The Sp5C contains a marginal zone (lamina 1), a substantia gelatinosa resembling Rexed’s lamina 2, and a magnocellular layer (laminae 3 and 4). The marginal zone consists of a thin sheet of cells containing large multipolar neurons, as well as small and medium-sized neurons. In rats, cats, and monkeys, this zone also includes fusiform and pyramidal neurons [1,35]. Research has shown that fusiform and pyramidal neurons correspond to nociceptive-specific and cold responses, respectively [1]. Inputs to the marginal zone come primarily from small-diameter myelinated fibers and unmyelinated afferents [36]. Many unmyelinated fibers contain substance P and CGRP, and some fibers contain pituitary adenylate cyclase-activating polypeptide (PACAP) [37]. Lamina 1 neurons project to several thalamic regions, including VPM, Po, the midline and intralaminar nuclei, and the posterior part of the ventromedial nucleus [38,39,40,41] (Figure 2). Lamina 1 also provides both direct hypothalamic projections [42] (Figure 2) and indirect projections through the parabrachial area [43,44]. The substantia gelatinosa (lamina 2) of the Sp5C consists of densely packed small, oval, or fusiform neurons and is rich in neuropeptides such as substance P, neurokinins A and B, CGRP, cholecystokinin, and somatostatin [45,46,47]. This region is particularly abundant in GABAergic somata and fibers [22,48]. The projections from lamina 2 are predominantly local to the adjacent magnocellular zone and the reticular formation in primates [49] (Figure 2). The magnocellular zone (laminae 3/4) contains medium-diameter oval or fusiform neurons interspersed with small and large neurons. Many neurons in this region contain glutamate, and some project to the VPM of the thalamus [1,20]. This lamina projects to the VPM, zona incerta, the facial nucleus, the trigeminal motor nucleus, adjacent reticular formation, and the ipsilateral spinal cord [1,20] (Figure 2). Magnocellular neurons also project to more rostral trigeminal nuclei, Sp5O and Sp5I in primates [49], cats [50], or rats [51].

The intersubnuclear connections within the Sp5 reciprocally exist between Sp5O and Sp5i [26,52,53,54], Sp5i and Sp5C [26,52,53,55,56], and Sp5C and Sp5O [26,52,53,54,55,57,58,59,60,61]. It has also been found that there are the abundant ascending and descending synaptic transmissions into each subnucleus of Sp5, with the exception of the substantia gelatinosa area and deep layer neurons of Sp5C from Sp5O [62]. Furthermore, the intersubnuclear connections of Sp5 are exclusively mediated by α-amino-3-hydroxy-5-methyl-4-isoxazolepropionic acid (AMPA) receptors and *N*-methyl-D-aspartate receptor (NMDA) receptors for excitatory transmission, and by GABA_A_ receptors and glycine receptors, or GABA_A_ receptors alone, for inhibitory transmission. It is suggested that three potential systems-the spinal trigeminal tract, the deep bundles, and the main body of Sp5-contribute to the intersubnuclear connections [26,52,53,54,60,63]. The main body of Sp5 is likely the most significant to connect subnuclei, particularly a subnucleus to the adjacent subnucleus by “chain-like” connections [52]. Additionally, there is a discrepancy in the distribution of central primary afferents among the subnuclei; central axons of neurons innervating tooth pulp terminate more heavily in Sp5O than Sp5C [23,24,64,65,66], while those innervating facial areas primarily to the Sp5C [67]. Thus, it is possible that the sensory information at the Sp5C area from the facial area can synaptically modulate the sensory information arriving at the Sp5O from the tooth pulp via the ascending pathways from Sp5C to Sp5O [60,68].

In addition to the intersubnuclear connections within the Sp5, there are reciprocal interactions between the Sp5 and autonomic parasympathetic nuclei. Notably, the Sp5 sends and receives information to and from the ipsilateral Edinger–Westphal nucleus [69,70] and the ipsilateral superior salivatory nucleus [71,72]. Furthermore, some neurons in the Sp5 are connected to the inferior salivatory nucleus [73].

#### 2.2.4. The Paratrigeminal Nucleus

The Pa5, originally identified as interstitial cells by Cajal, comprises groups of neurons located within the dorsal part of the spinal tract at the caudal level of Sp5I and the Sp5I/C boundary [1,20]. Neurons in this region are predominantly small and dispersed among the axons of the trigeminal tract. The Pa5 receives inputs from perioral and intraoral areas, such as the tooth pulp and cornea in rats [24], as well as from the upper gastrointestinal tract via trigeminal and glossopharyngeal afferents and possibly vagal afferents [1] (Figure 2). This region exhibits high immunoreactivity for CGRP and substance P [74]. Inflammation of the temporomandibular joint or chemical irritation of the tongue results in an increase in c-Fos and preprodynorphin in the Pa5 [75,76,77], suggesting that the Pa5 may serve as a crucial integrating area for nociceptive somato–visceral reflexes involving the upper gastrointestinal tract [78]. Subsequently, it was established that the integration of nociceptive inputs from the orofacial somatic area and the visceral organ, such as the lips and upper alimentary tract, respectively, is mediated by calbindin D28k-containing Pa5 neurons [79]. Furthermore, lesions in the Pa5 affect the mechanical threshold in the hind paw and nocifensive behaviors in rats [80], further indicating that the Pa5 plays a significant role in the central processing of mechanical and chemical nociceptive signals.

## 3. Synaptic Transmission and Plasticity in the Trigeminal Sensory System

Both excitatory and inhibitory synaptic transmissions are the two primary forms of communication between trigeminal primary afferent fibers and neurons in the Sp5, as well as among its subnuclei [62]. The balance between excitatory and inhibitory synaptic transmission within the nucleus is crucial for processing sensory information, including pain and temperature. In the Sp5, excitatory synapses can be formed by primary afferent fibers from the mandibular, maxillary, and ophthalmic regions of the face [81], as well as the intraoral structures such as teeth and the anterior two-thirds of the tongue, and by interneurons within the nucleus. Excitatory synaptic transmission involves the release of excitatory neurotransmitters, such as glutamate, which binds to its ionotropic receptors, including AMPA-, kainate-, and NMDA-type receptors, on the postsynaptic neuron. AMPA receptors are the primary mediators of fast excitatory synaptic transmission, as they are permeable to sodium ions, leading to depolarization of the postsynaptic membrane. The subunits of the AMPA receptor, such as GluA1 and GluA2/3, have been identified in the Sp5C [82]. Kainate receptors can mediate both fast and slow synaptic transmission by allowing the influx of sodium and calcium ions [83]. The trigeminal dorsal horn area expresses kainate receptor subunits, including GluK1, GluK2, and GluK3, which are postsynaptic to substance P-containing axon terminals [84]. NMDA receptors require both glutamate binding and membrane depolarization to open, permitting calcium ions to enter the synapse, which can lead to long-term changes in synaptic strength. Given the role of NMDA receptors, their activation is believed to be critical for the development and maintenance of central sensitization, a phenomenon in which the nervous system becomes hypersensitive to painful stimuli [85].

Inhibitory neurotransmitters in the Sp5 include GABA and glycine, which act on GABA_A_ receptors and glycine receptors, respectively, on the postsynaptic membrane. This action leads to hyperpolarization and a reduction in the neuronal firing rate due to the influx of chloride ions. Inhibitory synaptic transmission primarily originates from local interneurons within the Sp5 and plays a critical role in regulating the processing of sensory information from the orofacial region. Furthermore, inhibitory neurons in the trigeminal nucleus have been implicated in various neurological and psychiatric disorders. For instance, dysfunction in inhibitory synaptic transmission within the Sp5 has been associated with chronic pain conditions, such as trigeminal neuralgia and migraine headaches. Similarly, in other brain regions, abnormalities in inhibitory synaptic transmission have been linked to anxiety, depression, and schizophrenia.

Synaptic plasticity is a fundamental property of the brain that refers to the ability of synapses to modify their strength or effectiveness. It serves as the foundation for learning and memory formation, enabling neural circuits to adapt and reorganize in response to experiences and environmental changes. There are two primary types of activity-dependent synaptic plasticity: long-term potentiation (LTP) and long-term depression (LTD). LTP involves the strengthening of synaptic connections between neurons, resulting in enhanced communication [86], while LTD entails the weakening of these connections, leading to diminished communication. The balance between LTP and LTD is believed to be critical for normal brain function. Furthermore, synaptic plasticity plays a vital role in various neurological and psychiatric conditions, including Alzheimer’s disease, Parkinson’s disease, and depression.

In the Sp5, activity-dependent synaptic plasticity has been observed in the trigeminal afferent synapses as well as in the synapses between subnuclei of the Sp5. Notably, LTP was induced by high-frequency stimulation and was found to be independent of NMDA receptors; instead, it relied on metabotropic glutamate receptors (mGluRs), specifically mGluR1 and 5 [87], with a particular emphasis on mGluR5 [88]. This form of LTP within the pain transmission pathways has been proposed as a cellular mechanism underlying persistent pain [89]. In an ascending pathway from the Sp5C to the Sp5O in the Sp5 [60], which is a crucial transmission route for somatic sensation and nociception originating from sensory organs in the orofacial region [68,90,91], it has been demonstrated that LTP can also be induced in its excitatory transmission [92]. The LTP induced by 2 Hz stimulation of the Sp5C during the recording of depolarizing Sp5O neurons was dependent on NMDA receptors. In the same pathway, the activation of group I mGluRs using (*S*)-3,5-dihydroxyphenylglycine induced LTP that was independent on NMDA receptors [93]. This process involved Gq protein-coupled signaling molecules, L-type voltage-gated Ca^2+^ channels (VGCCs), and canonical TRP (TRPC) channels [94]. Furthermore, under conditions of reduced basal synaptic transmission, this LTP was mediated by N-type or P/Q-type VGCCs [95]. Conversely, in the ascending pathway from the Sp5C to the Sp5I, theta burst stimulation (5Hz) induced LTP of excitatory synaptic transmission, which was inhibited by the induction of LTP in GABAergic inhibitory synaptic transmission [96]. This GABAergic LTP was mediated by NMDA receptors and nitric oxide-cGMP signaling. Additionally, another type of LTP could also be induced in this pathway in an NMDA receptor-independent manner, ultimately mediated by mGluR1, but not mGluR5 [97]. Interestingly, a slow development of LTD at inhibitory GABAergic synapses coincided with the induction period of this LTP.

It has been suggested that the intersubnuclear ascending and descending excitatory and inhibitory transmissions in the trigeminal system [68] function as a multilevel filtering system to eliminate irrelevant sensory inputs. This process allows specific types of sensory information to pass through the first gate in the CNS [98]. Consequently, the LTP and LTD of glutamatergic and GABAergic inputs in the Sp5 neurons may play a crucial role in modulating, extracting, and integrating somatosensory signals from various orofacial sensory organs. Furthermore, given the presence of numerous nociceptive neurons in the trigeminal system [68], the LTP of excitatory transmission may also provide cellular mechanisms for amplifying nociceptive information within pain transmission pathways [89].

In the context of the trigeminal system, which is essential for orofacial pain, various forms of activity-dependent synaptic plasticity have been identified that contribute to the amplification or generation of pain signals. These phenomena include what are known as ‘wind-up’ and ‘central sensitization’. Wind-up refers to the progressive enhancement of neuronal responses during repeated stimulation of high-threshold sensory fibers, particularly C-fibers, as first described by Mendell and Wall [99]. This phenomenon is characterized by a homosynaptic mechanism, meaning it occurs at the same synapse where the stimulation is applied. In contrast, central sensitization involves an increased responsiveness of neurons following a period of repetitive stimulation, even after the stimulation has ceased. This phenomenon, as described by Woolf [85], is associated with a heterosynaptic mechanism, indicating that it involves interactions across different synapses. Studies suggest that wind-up may serve as an initial process leading to central sensitization [100,101,102,103]. Furthermore, the maintenance of central sensitization involves changes in microglia, astrocytes, gap junctions, membrane excitability, and gene transcription, highlighting a complex interplay of cellular and molecular mechanisms that sustain pain hypersensitivity [85]. Wind-up has been observed in convergent neurons of the Sp5O [104] and in the Sp5C [105]. It is suggested that wind-up, which plays a role in the temporal summation of trigeminal nociceptive responses, is crucial for understanding pain processing in the trigeminal system. This is particularly relevant as patients with painful temporomandibular disorders exhibit altered sensitivity to evoked pain, accompanied by changes in the temporal summation of pain [106]. Central sensitization is manifested in Sp5 neurons as an increase in spontaneous activity, an expansion of receptive field size, and heightened responsiveness to stimuli [107,108,109]. These changes have also been observed in the Sp5O nociceptive neurons following mustard oil application to the tooth pulp [110]. The central sensitization in Sp5O neurons is dependent on the activity of Sp5C neurons [90].

## 4. Orofacial Pain Related to the Trigeminal Sensory System

### 4.1. Orofacial Neuropathic Pain

The trigeminal nerve, which is responsible for sensation in the mouth, teeth, and face, is the largest cranial nerve and plays a crucial role in the perception of pain in the orofacial region. Injury or inflammation of this nerve can result in chronic neuropathic pain, characterized by spontaneous pain, hyperalgesia, and allodynia [111]. The underlying mechanisms contributing to the development and maintenance of orofacial neuropathic pain include peripheral and central sensitization, the involvement of ion channels and neurotransmitters, and neuroinflammation [112].

Peripheral sensitization plays a crucial role in the initiation of orofacial neuropathic pain [113]. Following injury or damage to the trigeminal nerve, primary afferents, or nociceptors in the affected area, become hyperexcitable [114]. This hyperexcitability results from various changes at the molecular and cellular levels, including the upregulation of ion channels, alterations in receptor function, and increased expression of pro-inflammatory mediators [115]. After nerve injury, there is an increased expression of voltage-gated sodium channels (VGSCs) (e.g., Nav1.7, Nav1.8, and Nav1.9) and VGCCs (e.g., Cav3.2) in the damaged neurons [116]. This upregulation leads to enhanced excitability and spontaneous ectopic discharges, contributing to the development of neuropathic pain triggered by nerve injury [117]. The function of receptors such as TRP channels, including TRPV1, which is involved in heat sensation and pain, is altered, resulting in heightened sensitivity to thermal and mechanical stimuli [118]. Additionally, injury to the trigeminal nerve leads to the release of pro-inflammatory cytokines (e.g., TNF-α, IL-1β, IL-6) and chemokines, which sensitize nociceptors and contribute to the development of peripheral sensitization [119].

Central sensitization plays a crucial role in the development and maintenance of chronic orofacial neuropathic pain [113]. In animal models of orofacial neuropathic pain, neurons in the trigeminal sensory system exhibit central sensitization, and these plastic changes result in an exaggerated response to stimuli and the persistence of chronic pain conditions [113,120]. This process involves hyperexcitability of central neurons, LTP, microglial activation, and a loss of inhibitory control [121,122]. Following peripheral nerve injury, the hyperexcitability of neurons in the spinal trigeminal nucleus and higher centers, such as the thalamus and cortex [123], is attributed to be caused by an increased release of excitatory neurotransmitters, including glutamate and substance P, alongside a decrease in inhibitory neurotransmission [119,124]. Central sensitization is linked to LTP of synaptic connections [92] and is also observed in cases of orofacial neuropathic pain. In the context of pain, LTP within the pain pathways results in prolonged and amplified pain signals [125]. Furthermore, injury to the trigeminal nerve activates microglia [126,127,128], and the activated microglia release pro-inflammatory cytokines, nitric oxide, and brain-derived neurotrophic factor (BDNF), which further enhance neuronal excitability and contribute to central sensitization [129]. A reduction in the function of inhibitory interneurons leads to decreased levels of the inhibitory neurotransmitters GABA and glycine, exacerbating the excitability of central neurons and perpetuating the pain state [128,130].

Third, ion channels and receptors, such as VGSCs, TRP channels, and NMDA receptors, play pivotal roles in the pathogenesis of orofacial neuropathic pain [131,132]. The upregulation of VGSCs in injured neurons increases neuronal excitability, and mutations in these channels, particularly in Nav1.7, have been linked to neuropathic pain disorders [133]. Previous studies have revealed that TRPV1 and TRPA1 are associated with the sensation of noxious heat and mechanical pain [134,135]. Alterations in the expression and function of these channels following nerve injury contribute to the heightened pain response [136]. NMDA receptors are crucial for the LTP of excitatory transmission in the Sp5O [92].

Neuroinflammation in orofacial neuropathic pain is closely associated with both peripheral and central neuroinflammation, which contribute to the pathogenesis of this condition [119]. Peripheral neuroinflammation involves the release of pro-inflammatory cytokines and chemokines from immune cells, glial cells, and neurons, creating a pro-inflammatory environment that sensitizes peripheral nociceptors and generates pain signals [137]. Central neuroinflammation occurs when microglia and astrocytes in the CNS become activated in response to peripheral nerve injury [138,139]. Activated glial cells release cytokines, chemokines, and other mediators that promote neuronal hyperexcitability and sustain central sensitization. This interaction between glial cells and neurons is a key driver of chronic pain [138].

Conclusively, orofacial neuropathic pain associated with the trigeminal sensory system is influenced by intricate mechanisms that involve peripheral and central sensitization, ion channel dysfunction, and neuroinflammation. Recent advancements in our understanding of these mechanisms have paved the way for novel therapeutic interventions. Ongoing research into the molecular foundations of this condition is crucial for developing more effective treatments and enhancing patient outcomes.

### 4.2. Trigeminal Neuralgia (Tic Douloureux)

Trigeminal neuralgia (TN) is defined by the International Association for the Study of Pain (IASP, 1994) as “sudden, usually unilateral, severe, brief, stabbing, recurrent pains in the distribution of one of more branches of the fifth cranial nerve”, and is particularly triggered by innocuous stimuli affecting the face or intraoral trigeminal territory [140]. Classic or typical TN can be classified as either idiopathic (primary; without identifiable structural lesions) or secondary to lesions such as tumors [140,141], cysts, or multiple sclerosis [140]. Another variant of TN, characterized by constant, burning, gnawing pain following the cessation of initial sharp pain, is classified as ‘TN with concomitant continuous pain,’ which was previously classified as atypical (type 2) TN [142]. It has become evident that a portion of seemingly idiopathic TN is caused by compression of the trigeminal roots by small, tortuous branches of the basilar artery, primarily the superior cerebellar artery and, occasionally, the anterior inferior cerebellar artery, in the posterior fossa [143,144]. However, similar vascular compression can also be observed in asymptomatic individuals. Conversely, no vascular compression of the nerve can be observed in approximately 25% of TN patients [140,145,146], with approximately half of these cases presumed to be due to secondary causes such as multiple sclerosis or tumors. Both vascular compression and conditions like multiple sclerosis and tumors lead to focal demyelination and injury of the trigeminal nerve [147,148]. This results in hyperexcitable axons, ectopic generation of impulses with high-frequency afterdischarges, and ephaptic (extrasynaptic) transmission of electrical impulses from low-threshold touch fibers to pain fibers due to the transmembrane passage of ions between axons [149]. Consequently, the paroxysmal pain experienced by TN patients can be attributed to the heightened susceptibility of low-threshold Aβ fibers to demyelination from mechanical damage or multiple sclerosis, as well as the ephatic transmission of ectopic impulses in Aβ fibers to pain fibers, which redirects light touch sensations to intense pain perception [140]. Additionally, evidence of pontine plaques, which damage primary trigeminal afferent fibers, has been found in multiple sclerosis patients with TN [150,151]. As a result, these patients often do not respond to surgical microvascular decompression surgery aimed at relieving the trigeminal nerve from vascular compression. In TN cases, involvement of the V2 or V3 branches is common (with only one branch affected in approximately 60% of cases; both branches in approximately 35%), while the participation of the V1 branch is rarely involved (~4%) [152]. Typically, TN patients can identify the trigger zone and report specific activities that provoke an attack. Furthermore, the incidence of TN is higher in women, with a female-to-male ratio of approximately 2–3:1 [153,154], and attacks can occur 10 to 100 times per day in most patients [144].

The underlying mechanisms of TN remain poorly understood. Arterial compression of the trigeminal nerve and its focal demyelination are the primary causes of TN. Numerous molecular mechanisms associated with TN have been identified, particularly changes in sodium and potassium channels within the trigeminal ganglion neurons. For instance, a rat model of TN induced by infraorbital nerve constriction demonstrated an upregulation of Nav1.3 and a downregulation of Nav1.7 [155,156], as well as an upregulation of Kv7.2 channels, which generate M-currents (muscarinic potassium currents), in the trigeminal ganglion neurons [157,158]. Additionally, a mutation in the SCN8A gene, which encodes Nav1.6, was identified in a patient with TN [159].

The ectopic discharge and ephaptic transmission induced by axonal demyelination gradually lead to the development of central sensitization, resulting in background pain that fluctuates and is described as burning, throbbing, or aching. Consequently, if the initial paroxysmal pain is not alleviated by effective medication, it may evolve into a different form, such as TN characterized by continuous pain [140]. Conversely, the stimulation of Sp5 can elicit TN-like responses. The injection of epileptogenic agents into the Sp5 of cats and monkeys has been shown to induce pain syndrome [160], and irritation of the descending tract of the trigeminal nerve, as well as the Sp5, enhances the response to electrical stimulation in the peripheral portion of the nerve [161]. Notably, a pontine infarction caused by an arterial embolism resulted in TN in a human patient [162]. These findings suggest that the trigeminal sensory nuclei, particularly the Sp5, along with the trigeminal ganglion, play a significant role in mediating TN. In conclusion, alteration in the pain transmission circuits of the Sp5 and neuroplastic changes within the Sp5 may underlie continuous or long-lasting pain between paroxysmal attacks.

### 4.3. Headache

Headache disorders, a significant subset of orofacial pain conditions, are closely intertwined with the trigeminal sensory system. These disorders are broadly categorized into primary headaches, such as migraine, tension-type headache (TTH), and trigeminal autonomic cephalalgias (TACs), and secondary headaches, which are attributed to underlying pathologies.

Migraine is a prevalent primary headache disorder characterized by recurrent episodes of moderate to severe, often unilateral, pulsatile headaches. The pathophysiology of migraine involves complex interactions within the trigeminovascular system. The activation of trigeminal nociceptors triggers the release of vasoactive neuropeptides, particularly calcitonin gene-related peptide (CGRP), at the dura mater, which subsequently induces neurogenic inflammation and sensitizes both peripheral and central pain pathways [163]. During a migraine attack, the trigeminal ganglion releases CGRP, substance P, and PACAP, which contribute to vasodilation and the subsequent sensitization of trigeminal neurons, thereby intensifying the pain [164]. Central sensitization plays a crucial role in the chronification of migraine. Persistent nociceptive input induces neuroplastic changes in the Sp5C and higher brain centers, such as the thalamus, resulting in a lowered pain threshold and increased susceptibility to headache attacks [165]. Cutaneous allodynia, experienced during migraine episodes, serves as a clinical marker of this central sensitization [166].

TTH represents the most common primary headache disorder characterized by bilateral, non-pulsating pain that is generally unaffected by physical exertion. In TTH, central sensitization of the trigeminal nerve plays a particular significant role, especially in chronic cases. Prolonged nociceptive input from pericranial myofascial tissues heightens the responsiveness of second-order neurons within the trigeminal nucleus to low-threshold mechanosensitive afferents, contributing to the persistence and severity of TTH [167].

TACs represent a group of disorders characterized by recurrent, severe unilateral pain in the trigeminal distribution, accompanied by autonomic symptoms such as facial flushing, lacrimation, and nasal congestion. The pathophysiology of TACs is centered around the trigeminal-autonomic reflex, where activation of the trigeminal nerve induces parasympathetic outflow through the superior salivatory nucleus (SSN) and the sphenopalatine ganglion (SPG), resulting in the characteristic autonomic symptoms [168].

Recent research has highlighted the hypothalamus’s role in headache disorders, particularly in TACs and migraine. Functional neuroimaging studies have shown hypothalamic activation during spontaneous migraine headaches and the premonitory phase, suggesting its involvement in both nociceptive and autonomic regulation [169]. The hypothalamus connects to the trigeminovascular system and influences sympathetic and parasympathetic neurons in the brainstem, thereby contributing to headache pathology [170]. The thalamus serves as a critical hub in headache disorders, receiving inputs from second-order trigeminovascular neurons and projecting to cortical regions involved in autonomic, affective, and cognitive functions [171]. Thalamic dysfunction has been implicated in central sensitization, photophobia, and allodynia related to migraine [165].

Cortical involvement in headache disorders is particularly evident in migraine with aura. The aura phenomenon is believed to be a manifestation of cortical spreading depression (CSD), a wave of neuronal and glial depolarization followed by a period of neuronal quiescence. While CSD can activate the trigeminovascular system in animal models, its precise role in human migraine pathophysiology remains an area of ongoing research [172].

Understanding these complex mechanisms has led to the development of novel therapeutic approaches. CGRP-targeted therapies, including gepants for acute treatment and monoclonal antibodies for prevention, have shown promising results in clinical trials [173]. Neuromodulation techniques, such as single-pulse transcranial magnetic stimulation (sTMS) and non-invasive vagus nerve stimulation (nVNS), are emerging as effective therapeutic options for modulating trigeminal nociceptive pathways [174].

In conclusion, the trigeminal sensory system plays a crucial role in the pathophysiology of various headache disorders. The intricate interactions between the peripheral and central components of this system, along with the involvement of key brain regions such as the Sp5C, the hypothalamus, and the thalamus, contribute to the diverse clinical presentations of these conditions. Ongoing research is continually uncovering these mechanisms, paving the way for the development of more targeted and effective treatments for headache-related orofacial pain.

### 4.4. Burning Mouth Syndrome

One of the primary concerns in chronic orofacial pain is burning mouth syndrome (BMS). This condition is a chronic orofacial pain condition characterized by a burning or dysaesthetic sensation in the mouth without visible lesions, as defined by the International Headache Society [175]. Symptoms, which include irritation, burning, tingling, itching, or numbness, must persist for at least two hours a day over a period of three months. Some patients also report experiencing dry mouth and changes in taste [176,177]. BMS are typically moderate to severe and often bilateral. The prevalence of BMS ranges from 0.1% to 3.9%, with higher rates observed in postmenopausal women aged 50 to 70 [178,179].

The exact cause of BMS remains unclear; however, it is believed to arise from a combination of peripheral, central, and psychological factors [180]. Recent neurophysiological research highlights BMS as a neuropathic pain condition characterized by dysfunction in both peripheral and central nervous pathways [181]. Some studies point to peripheral dysfunction [182,183,184,185,186,187], while others propose that alterations in the central organization of pain perception occur at multiple levels [188].

The high prevalence of oral parafunctional habits and dry mouth symptoms in BMS patients suggests that traumatic irritation to the oral mucosa may lead to neuropathic changes [189]. Tongue mucosal biopsies from BMS patients show a significant reduction in intraepithelial nerve fiber density compared to healthy controls, indicating focal sensory neuropathy of trigeminal small fibers and atrophy of oral mucosa [190,191]. Immunohistochemical staining of these biopsies reveals increased levels of nociceptive molecules such as TRPV1, NGF, and P2X3 [191,192]. Consequently, BMS patients exhibit a decreased tolerance to noxious heat stimulation [193]. Furthermore, BMS patients show reduced expression of CB1 receptors and increased expression of CB2 receptors in their tongues, suggesting a role for TRPV1 in hypersensitivity and dysaesthesia [193,194]. P2X3 receptors, activated by ATP, contribute to burning pain [195], while non-neuronal cells synthesize NGF, which modulates peripheral pain [196]. The elevated level of NGF may be associated with the increased insertion of TRPV1 on the membrane of sensory neurons or nerves [197], contributing to peripheral neuropathy in BMS [191,196]. Studies utilizing lingual nerve blocks with lidocaine and benzocaine application showed mixed results (decreasing, no change, increasing) when assessed using visual analog scale (VAS) scores, indicating the involvement of both peripheral and central neuropathic mechanisms in BMS [198,199].

BMS is an unusual idiopathic condition where pain and taste disturbances are often linked. Patients frequently report taste disturbances like bitter or metallic phantom tastes and dysgeusia, despite the absence of visible lesions [182,183,184,185]. These symptoms may result from nerve damage carrying gustatory information, particularly the chorda tympani nerve [200], a branch of the facial nerve (Figure 1). Damage to this nerve can cause abnormal pain or taste sensations. Studies suggest that greater taste loss correlates with increased oral pain, implicating chorda tympani nerve hypofunction in BMS pathophysiology [200,201,202]. Furthermore, damage to peripheral gustatory fibers may affect central inhibitory mechanisms, leading to trigeminal hypersensitivity and dysgeusia [201]. BMS patients often have a higher density of fungiform papillae and show reduced gustatory sensitivity but increased pain thresholds for capsaicin-induced burning pain. This supports the hypothesis that damage to peripheral taste nerve fibers contributes to the pain experienced in BMS. Taste dysfunction may indicate disorders of small afferent nerve fibers [184,203].

BMS patients exhibit a variety of symptoms individually; however, quantitative sensory testing (QST) studies have demonstrated that hypofunction of Aδ fibers, mediating innocuous cooling, is more prevalent than hypofunction of C fibers [204]. This suggests that the loss of Aδ fiber-mediated (cool) inhibition on C fibers, which are responsible for heat sensation, may lead to the development of persistent burning sensations [204]. Another hypothesis posits that damage to the Aδ taste afferent fibers, which travel along the chorda tympani nerve, could contribute to pain associated with BMS. An electro-gustatometry study further revealed hypofunction of Aδ fibers in the chorda tympani, as evidenced by elevated taste detection thresholds in the tongue mucosa of BMS patients [198,205]. The reduction in inhibitory control of taste fibers over afferent stimuli may result in burning pain.

Lastly, animal models of BMS are essential for gaining a better understanding of the underlying mechanisms of BMS. However, due to incomplete knowledge of BMS pathogenesis, there are currently no experimental animal models that have been integrated into BMS studies.

## 5. Concluding Remark

Sensory and pain signals from orofacial area including tooth and cornea are processed in the trigeminal sensory nuclei in the brainstem. This review discusses the cellular composition, connections, and synaptic transmission in the nuclei, the extra-trigeminal influences including parasympathetic flow, and four representative types of orofacial pain. The information further contributes to the fundamental understandings of orofacial pains.

## Figures and Tables

**Figure 1 ijms-25-11306-f001:**
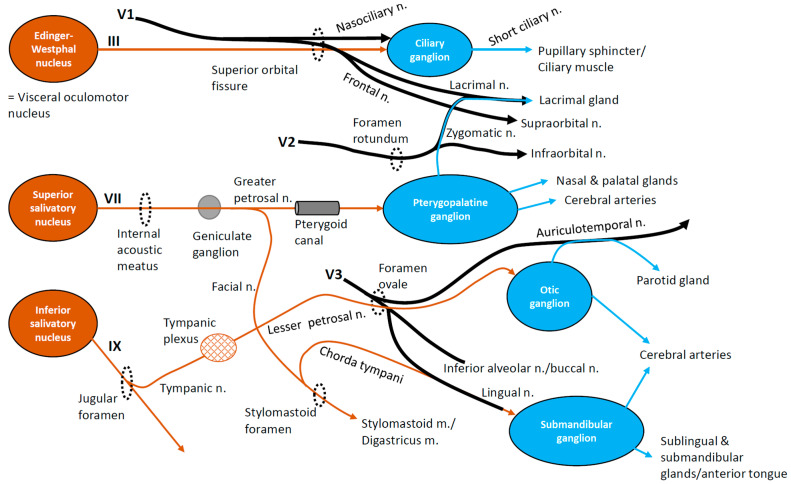
The convergence of trigeminal nerves and autonomic parasympathetic nerves. V1, ophthalmic nerve; V2, maxillary nerve; V3, mandibular nerve; III, oculomotor nerve; VII, facial nerve; IX, glossopharyngeal nerve. Black color, trigeminal nerve branches; brown, preganglionic parasympathetic; light blue, postganglionic parasympathetic.

**Figure 2 ijms-25-11306-f002:**
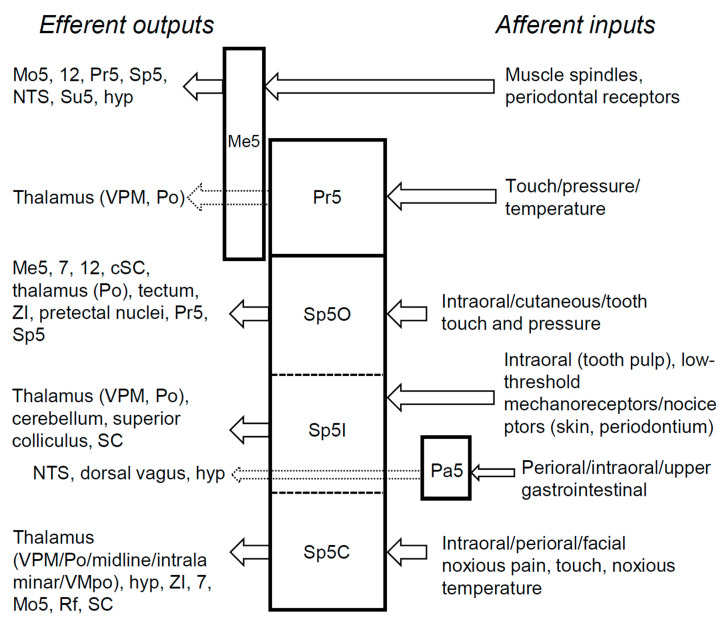
Selected primary afferent inputs and efferent outputs of the nuclei in the trigeminal sensory system. Mo5, trigeminal motor nucleus; 7, facial nucleus; 12, hypoglossal nucleus; Sp5, spinal trigeminal nucleus; NTS, nucleus of tractus solitarius; Su5, supratrigeminal nucleus; hyp, hypothalamus; VPM, ventroposteromedial nucleus; Po, posterior nucleus; ZI, zona incerta; Pr5, principle nucleus; SC, spinal cord; cSC, cervical spinal cord; Rf, reticular formation.

## Data Availability

No new data were created or analyzed in this study. Data sharing is not applicable to this article.

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
