# Peer review of "The Trigeminal Sensory System and Orofacial Pain"

_ijms, 2024, doi:10.3390/ijms252011306_

Round 1

Reviewer 1 Report

Comments and Suggestions for Authors

This review paper summarizes the recent advances in our knowledge of the trigeminal sensory system. It describes in detail the trigeminal sensory nuclear complex with particular emphasis on the spinal trigeminal nucleus that primarily relays sensory information related to pain from the oral cavity and face. The authors also report the basic mechanisms and four main types of orofacial pain. The information provided is interesting and will be beneficial for researchers dealing with the trigeminal sensory system and its clinical significance for primary headaches. Specifically, relatively few issues merit comment and there are some generally minor details to be settled:

General comments:

  1. The authors should pay attention to the fact that the mesencephalic trigeminal nucleus is localized not only in the midbrain; it extends from the upper midbrain to the mid-pontine levels where most of its neurons are situated.
  2. The fourth sentence in the second paragraph on p. 2 (lines 55-56) should be clarified because it is not fully understandable.
  3. Addition of one closing paragraph with a few short concluding sentences will provide the readers with what a functional concept can be drawn from the observations.

Minor concerns:

  1. Page 2, line 72, “providing innervating the cheek ” should be “providing innervation of …”
  2. Page 3, Figure 1, the term “lessor petrosal n.” should be replaced with “lesser …”.
  3. Throughout the text, better use “principal sensory/mesencephalic trigeminal nucleus” or “principal sensory/mesencephalic nucleus of trigeminal nerve” rather than simply “mesencephalic/principal sensory

Paying attention to the above points will certainly improve the quality of this manuscript. 

Comments on the Quality of English Language

From a linguistic point, the writing is not bad, although a few minor errors are noted throughout the text. A more meticulous re-reading with regard of the grammatical structure of English will undoubtedly sort them out.

Author Response

Comment 1: The authors should pay attention to the fact that the mesencephalic trigeminal nucleus is localized not only in the midbrain; it extends from the upper midbrain to the mid-pontine levels where most of its neurons are situated.

Response 1: We thank the reviewer for this comment. We revised the part of Me5 by inserting the first sentence on p. 3 and, also, revised Fig. 2.

Comment 2: The fourth sentence in the second paragraph on p. 2 (lines 55-56) should be clarified because it is not fully understandable.

Response 2: We revised the sentence for clarification on p. 2.

Comment 3: Addition of one closing paragraph with a few short concluding sentences will provide the readers with what a functional concept can be drawn from the observations.

Response 3: We inserted ‘concluding remark’ in 5 on p. 13.

Minor comment 1: Page 2, line 72, “providing innervating the cheek ” should be “providing innervation of …” -> Response 1: We revised as suggested.

Minor comment 2: Page 3, Figure 1, the term “lessor petrosal n.” should be replaced with “lesser …”. Page 3, Figure 1, the term “lessor petrosal n.” should be replaced with “lesser …”.

-> Response 2: We revised it in the fig. 1.

Minor comment 3: Throughout the text, better use “principal sensory/mesencephalic trigeminal nucleus” or “principal sensory/mesencephalic nucleus of trigeminal nerve” rather than simply “mesencephalic/principal sensory

-> Response 3: We think that the text ‘of trigeminal nerve’ may not necessary; however, the text was change to abbreviation, e.g., Pr5 on p. 5.

Reviewer 2 Report

Comments and Suggestions for Authors The manuscript comprehensively describes the function of the trigeminal
system, and the connectivity with various nervous structures. The trigeminal sensory system is fundamental for the etiological
understanding and treatment of many painful syndromes. Those covered in the manuscript represent the main ones for which
knowledge of the functioning of the trigeminal system allows you
to choose the most appropriate therapeutic strategies.

Author Response

Comment 1: The manuscript comprehensively describes the function of the trigeminal
system, and the connectivity with various nervous structures. The trigeminal sensory system is fundamental for the etiological understanding and treatment of many painful syndromes. Those covered in the manuscript represent the main ones for which knowledge of the functioning of the trigeminal system allows you to choose the most appropriate therapeutic strategies.

Response 1: We have no response to reviewer #2.

Reviewer 3 Report

Comments and Suggestions for Authors

Dear authors, congratulations on this mighty work, however, some improvements are needed before it can be reconsidered for publication.

1) introduce some tables in the main sections so as to increase readability

2) trigeminal neuralgia, followed by trigeminal schwannoma is undoubtedly the first entity not only afflicting the ganglion. It might be interesting to discuss it especially in terms of surgical management I suggest two papers for you to read and cite: https://doi.org/10.3390/jcm13133701 ; https://doi.org/10.3390/jcm13092712

3) very interesting sections on various related disorders, I also point out sleep disorders in this regard you should read and cite: https://doi.org/10.3390/medsci12010002

I look forward to reading the revised manuscript.

Comments on the Quality of English Language

minor editing needed 

Author Response

Comment 1: introduce some tables in the main sections so as to increase readability

Response 1: We tried to figure out whether any more figures or tables are needed, but we think no more tables are necessary. If the reviewer #3 suggests a table, we will consider to insert it.

Comment 2: trigeminal neuralgia, followed by trigeminal schwannoma is undoubtedly the first entity not only afflicting the ganglion. It might be interesting to discuss it especially in terms of surgical management I suggest two papers for you to read and cite: https://doi.org/10.3390/jcm13133701 ; https://doi.org/10.3390/jcm13092712

Response 2: We inserted one of suggested references in [141].

Comment 3:  very interesting sections on various related disorders, I also point out sleep disorders in this regard you should read and cite: https://doi.org/10.3390/medsci12010002

Response 3: We think that this reference is not related to our review.

Round 2

Reviewer 3 Report

Comments and Suggestions for Authors

Dear authors, now the work is much better, it is readable, well structured and well organized into sections. Fix a few typos along the manuscript and alphabetize keywords, I suggest you think about it a bit and add more, as they are also the way through which your article can be found. 

Congratulations.